# Investigation of Laser Polishing of Four Selective Laser Melting Alloy Samples

**Dongqi Zhang [1,2], Jie Yu [1,2], Hui Li [1,2,3,*], Xin Zhou [4,*], Changhui Song [5,*], Chen Zhang [2], Shengnan Shen [1,2,3], Linqing Liu [5] and Chengyuan Dai [6]**

1.  School of Power and Mechanical Engineering, Wuhan University, Wuhan 430072, China; zhang_dongqi@whu.edu.cn (D.Z.); 2017202080005@whu.edu.cn (J.Y.); shen_shengnan@whu.edu.cn (S.S.)
2.  Research Institute of Wuhan University in Shenzhen, Shenzhen 518057, China; zhangchentm@163.com
3.  Key Laboratory of Hydraulic Machinery Transients (Wuhan University), Ministry of Education, Wuhan 430072, China
4.  Science and Technology on Plasma Dynamics Laboratory, Air Force Engineering University, Xi'an 710038, China
5.  School of Mechanical & Automotive Engineering, South China University of Technology, Guangzhou 510641, China; 201821002839@mail.scut.edu.cn
6.  Mechanical and Aerospace Engineering, University of California, San Diego, CA 92093, USA; dcyuan1208@gmail.com
*   Correspondence: li_hui@whu.edu.cn (H.L.); dr_zhouxin@126.com (X.Z.); song_changhui@163.com (C.S.); Tel.: +86-158-0273-5936 (H.L.); +86-138-9283-9739 (X.Z.); +86-189-8882-4667 (C.S.)



**Featured Application: This paper presents the in-situ laser polishing of four typical alloys in the selective laser melting (SLM) process.**

**Abstract:** Selective laser melting (SLM) is a layer by layer process of melting and solidifying of metal powders. The surface quality of the previous layer directly affects the uniformity of the next layer. If the surface roughness value of the previous layer is large, there is the possibility of not being able to complete the layering process such that the entire process has to be abandoned. At least, it may result in long term durability problem and the inhomogeneity, may even make the processed structure not be able to be predicted. In the present study, the ability of a fiber laser to in-situ polish the rough surfaces of four typical additive-manufactured alloys, namely, Ti6Al4V, AlSi10Mg, 316L and IN718 was demonstrated. The results revealed that the surface roughness of the as-received alloys could be reduced to about 3 μm through the application of the laser-polishing process, and the initial surfaces had roughness values of 8.80–16.64 μm. Meanwhile, for a given energy density, a higher laser power produced a laser-polishing effect that was often more obvious, with the surface roughness decreasing with an increase in the laser power. Further, the polishing strategy will be optimized by simulation in our following study.

**Keywords:** additive manufacturing; selective laser melting; laser polishing; typical alloys; surface roughness

## 1. Introduction

Additive manufacturing (AM) technology is regarded as one of the key technologies that will lead the 'third industrial revolution'. It is highly regarded in countries around the world and has been the subject of a large number of studies. In particular, selective laser melting (SLM) technology has attracted considerable attention. Although the process offers many advantages compared to traditional techniques, the surface roughness produced by the SLM technology limits its development

and potential high-end applications. Wang et al. concluded that the $R_a$ value of the parts fabricated by traditional mechanical methods such as milling and grinding is less than 1–2 µm, and the $R_a$ value of SLM parts is usually between 10 µm and 30 µm [1]. Mumtaz and Hopkinson pointed out that it is critical to obtain the required part top surface roughness in some applications [2]. However, there are many reasons for a rough surface. Many researchers have studied the effect of the SLM process parameters on the surface roughness. Hong et al. concluded that laser process parameters have crucial effects on the surface quality of SLM-fabricated Co–Cr dental alloys [3]. Wu et al. studied the influence of the laser power and laser scanning speed on the surface morphology [4]. Their results showed that, when the scanning speed was increased, the surface morphology initially became flatter, but then, roughness developed again as the scanning speed increased further. As the laser power was increased, the surface morphology gradually became rougher. Zhang et al. studied Ti6Al4V single-track, multi-track and bulk samples formed by SLM using different scanning speeds [5]. They found out that the $R_a$ of the top of the bulk increased with the scanning speed. Fox et al. analyzed the effect of the beam power, beam velocity and overhang angle in an attempt to further the understanding of the relationship between individual surface characteristics and surface roughness parameters [6]. Pupo used a full factorial design to assess the impact of process parameters on surface quality (Q), flatness, overlapping and surface roughness [7]. The best results were obtained with laser powers of 300 W and 400 W and scanning ranges of 450 µm and 600 µm. Cherry et al. investigated the relationship between the laser energy density and the properties of 316 L stainless steel [8], such as its porosity, surface roughness, microstructure, density and hardness. Koutiri et al. focused on adjusting the volume energy density with the goal of finding a compromise between an optimum densification state and a minimum amount of contaminating spatter [9], and they finally found an optimum set of parameters that delivered the best surface roughness, combined with low porosity. Khanna et al. found that the surface roughness was linearly related to the energy density in the process [10]. Wang et al. studied the effects of the laser energy density (LED) on the densities and surface roughness values of AlSi10Mg samples processed by SLM [11]. The results showed that the LED has an important influence on the surface morphology of the forming part, with a higher LED possibly producing a balling effect, and a lower LED tending to produce defects, such as porosity and microcracks, subsequently affecting the surface roughness and porosity of the parts. Yakout et al. and Shen et al. used a LED formula, as follows [12,13]:

$$E_v = \frac{P}{v * h * t} \qquad (1)$$

where $P$ is the laser power (W), $v$ is the scanning speed (mm/s), $h$ is the hatch space (mm), $t$ is the layer thickness (mm) and $E_v$ is the volumetric LED (J/mm$^3$). The LED directly determines the thermal input to the sample's surface and plays an important role in laser polishing.

Furthermore, there are other factors that affect surface quality. Zhang et al. studied the evolution of the molten pool by applying a multiphysics simulation and experiments and analyzed the influence of the flow field on the surface morphology of a single track [14]. In addition, Spierings et al. undertook an investigation on the effect of three different powder granulations on the resulting part density, surface quality and mechanical properties of the resulting materials [15]. They were able to attain a low surface roughness by optimizing the powder material. Liu et al. drew a conclusion that spatter, including powder spatter and droplet spatter, affected the SLM process and increased the surface roughness [16]. Esmaeilizadeh et al. studied the effects of powder spatter on the quality of parts at different positions on the manufacturing board [17]. The results showed that the surface roughness increased from 14.4 µm to 28 µm in the spatter-intensive areas of the board. Therefore, it is recommended that this area should not be used in the manufacture of high-quality parts. With a goal of better understanding these spatter phenomena, Zhang et al. used different technologies to prepare the powder so as to not only increase laser absorptivity but also greatly reduce the surface spheroidization phenomenon which can easily occur in metal powder printing [18], thus reducing the surface roughness. Kim et al. presented a systematic approach to improve the surface profile of AM parts using a computational model and

a multi-objective optimization technique [19]. All of the above studies aimed to optimize the SLM process parameters applied to the raw materials and printing process to obtain a smoother surface. However, the final product surface still exhibits a significant degree of roughness, which greatly affects the development of AM technology.

In recent years, much research has addressed the post-processing of SLM products to obtain a smooth surface. Chen et al. described a method to control the surface deformation during the SLM and found that increasing the magnetic field within a certain range decreased the impact of the Marangoni effect, resulting in a smoother surface [20]. Yu et al. sand-blasted on the top and bottom surfaces of the sample in order to obtain lower surface roughness [21]. Zhang et al. applied electrochemical polishing (ECP) to Inconel 718 components fabricated by selective laser melting, significantly reducing the surface roughness as a result [22]. Tyagi et al. demonstrated the utilization of a chemical polishing approach to improve the surface finish of as-produced metal additive manufacturing components [23]. Another post-processing method involves laser polishing. Ma et al. hold that the method was mainly based on the melting caused by the thermal input of laser irradiation [24], accompanied by the solid–liquid–gas phase change. Yung et al. presented a novel method for reducing the surface roughness of cobalt chromium (CoCr) components with a complex surface geometry by applying a polishing method [25]. They achieved a reduction in the surface roughness of up to 93% relative to the as-received samples. Ukar et al. carried out experimental studies to determine the influence of the different laser sources on the roughness reduction rate in laser polishing [26]. Lamikiz et al. presented a laser-polishing process for metallic sintered parts and the measured reductions were up to 80% reductions in $R_a$ parameter [27]. Mai et al. investigated laser polishing of 304 stainless steel using different process parameters such as laser output power, off-focus position, pulse frequency, scanning speed, and scanning strategy [28]. The present study focused on the influence of different laser-polishing strategies on the surface roughness of selective laser-melted samples. Typical alloys, commonly used in AM, namely, Ti6Al4V, AlSi10Mg, 316L and IN718, were studied. In addition, a 3D optical profiler was used to measure the surface roughness of the samples before and after polishing. The data were processed and analyzed using visualization software, and the effects of the laser power, scanning speed and LED on the surface roughness were studied. Finally, optimized laser-polishing parameters were obtained, which can be used to provide guidance in industrial applications and scientific research.

## 2. Materials and Methods

### 2.1. Sample Fabrication

Four typical alloys were selected in the present study, namely, Ti6Al4V, AlSi10Mg, 316L and IN718. The samples, all of which measured 40 mm × 40 mm × 5 mm, were fabricated using SLM equipment (SLM 125, SLM Solutions NA, Inc., Lübeck, Germany). The square plate-shaped samples were printed at the center of the build plate. The structure of the samples was simple and the height was very low, so the position may have a little influence on the global geometry. Therefore, a systematic compensation approach from forming technology presented by Hartmann et al. [29] has not been considered. The particle diameter of the metal powders (Advanced Powders and Coatings Inc., Boisbriand, Québec, Canada) used in the present study was 15–45 μm. Figure 1 shows scanning electron microscope (SEM, FEI Inspect F50, Thermo Fisher Scientific Inc., Waltham, MA, USA) images of the particles of the four typical alloy powders, showing the particle size range and that most of the powder particles are spherical.

Each of the four kinds of powders exhibits a good size distribution and surface roughness. Sutton et al. confirmed that these powder characteristics have a great impact on the performance of the samples [30], such as their mechanical strength, porosity and surface finish. As such, the virgin powder that was selected in the experiment can greatly reduce printing problems caused by powder quality, including oxidation and particle adhesion. All of the above factors affect the surface quality and mechanical properties of the printed parts and, sometimes, could lead to printing failures. In fact,

powders are often reused. Although they are screened, the powder quality will degrade during the recycling.

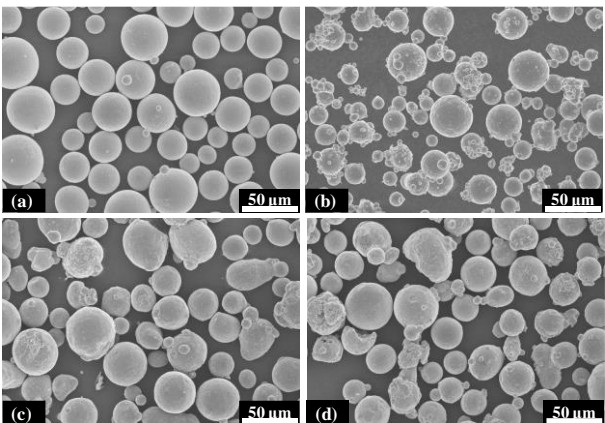

**Figure 1.** SEM images of four powder particles: (**a**) Ti6Al4V, (**b**) AlSi10Mg, (**c**) 316L and (**d**) IN718.

A continuous wave fiber laser (IPG YLR-400, IPG Photonics Corporation, Oxford, MA, USA) with a wavelength of 1060 nm, a nominal output power of 400 W and a spot size of 100 μm was used in the sample fabrication process. The specific printing process parameters are optimized and they are empirical values of the SLM equipment, shown as Table 1. After the printing was completed, the samples were not taken out of the chamber. Rather, a laser-polishing experiment was performed under an argon atmosphere with an oxygen content of <0.1%, which was the same as that of the printing processing environment, in which almost no oxidation occurs. The experimental setup used to perform the polishing experiments is shown in Figure 2.

**Table 1.** SLM parameters of four typical alloys.

|  | Ti6Al4V | AlSi10Mg | 316L | IN718 |
|---|---|---|---|---|
| Power (W) | 276 | 350 | 320 | 175 |
| Feed rate (m/s) | 0.76 | 1.15 | 0.65 | 0.60 |
| Layer thickness (μm) | 50 | 50 | 50 | 50 |
| Hatch spacing (μm) | 120 | 170 | 140 | 140 |

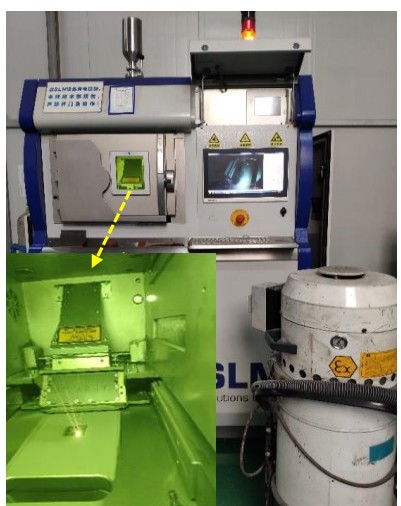

**Figure 2.** Experimental setup.

The sample surface was divided into nine areas, each measuring 10 mm × 10 mm. The laser-polishing experiment was performed using the same laser equipment but laser parameters were adjusted for laser polishing. The experimental parameters corresponding to eight regions are listed in Table 2. In addition, the scanning path of the laser is the same as that of the printing process, the hatch spaces of the 316L and IN718 alloys are adjusted to 80 μm, and the others remained unchanged. Subsequently, the sample was cut from the build plate using electrospark wire-electrode cutting equipment (FH-020C, Suzhou Xingjie CNC Technology Co., Ltd., Suzhou, China). The initial and post-processed surfaces of the four typical alloys are shown in Figure 3.

**Table 2.** Laser-polishing parameters for eight areas.

| Area Index | 1 | 2 | 3 | 4 | 5 | 6 | 7 | 8 | 9 |
|---|---|---|---|---|---|---|---|---|---|
| Power (W) | 100 | 100 | 200 | 200 | No | 300 | 300 | 400 | 400 |
| Feed rate (m/s) | 0.5 | 1.0 | 0.5 | 1.0 | No | 0.5 | 1.0 | 0.5 | 1.0 |

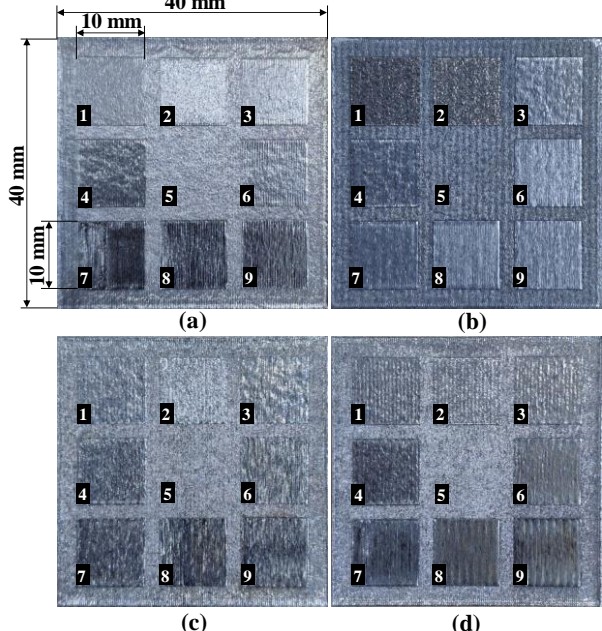

**Figure 3.** Initial and polished surfaces: (**a**) Ti6Al4V, (**b**) AlSi10Mg, (**c**) 316L and (**d**) IN718.

### 2.2. Morphology Observation by 3D Optical Profiler

Roughness measuring methods include laser reflectivity, contact stylus tracing, tactile profile measurement, focus variation, fringe projection technique and confocal laser scanning microscope [31,32]. Sun et al. presented a novel method based on convolutional neural networks (CNN) for making intelligent surface roughness identifications and achieved high-precision surface roughness estimation [33]. Patel et al. introduced a SRAS system capable of detecting surface ultrasound waves on the rough-surface of an as-deposited SLM sample [34]. In this work, the surface morphologies of the samples before and after laser polishing were observed using a 3D optical profiler (RTEC Up Dual-Mode, Rtec Instruments, San Jose, CA, USA). It was used to scan the nine areas to get nine sets of data, which were then processed using the Gwyddion free and open-source software, which is a modular programme for scanning probe microscopy (SPM) data visualization and analysis. Gadelmawla et al. expressed the line roughness of the samples in terms of its arithmetic mean height ($R_a$) [35]. $S_a$ is the extension of $R_a$ to a surface. It expresses, as an absolute value, the difference in

height of each point compared to the arithmetical mean of the surface. It is defined by ISO standard (ISO 25178-2:2012). The expressions of $R_a$ and $S_a$ are as follows:

$$R_a = \frac{1}{l} \int_0^l |y(x)| dx \tag{2}$$

$$S_a = \frac{1}{A} \iint_A |Z(x, y)| dx dy \tag{3}$$

where $l$ is the length of the surface profile, $y(x)$ is the deviation of the surface profile at a point $x$ from the mean height over the profile, $A$ implies that the integration is performed over the area of measurement and $Z(x, y)$ is the function representing the height of the surface relative to the best fitting plane, cylinder or sphere.

## 3. Results and Discussion

### 3.1. Characterisation of Surface Morphology

The three-dimensional contours of the nine regions were obtained using a self-programming code. Figures 4–7 show the surface morphologies of the Ti6Al4V, AlSi10Mg, 316L and IN718 samples, respectively.

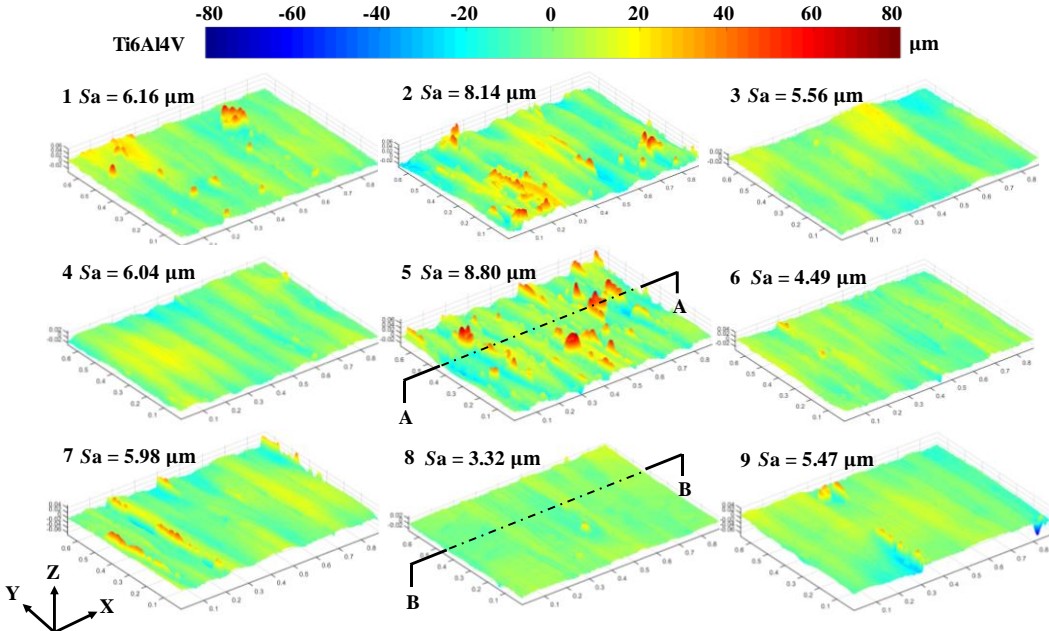

**Figure 4.** Three-dimensional contours of the Ti6Al4V sample after laser polishing.

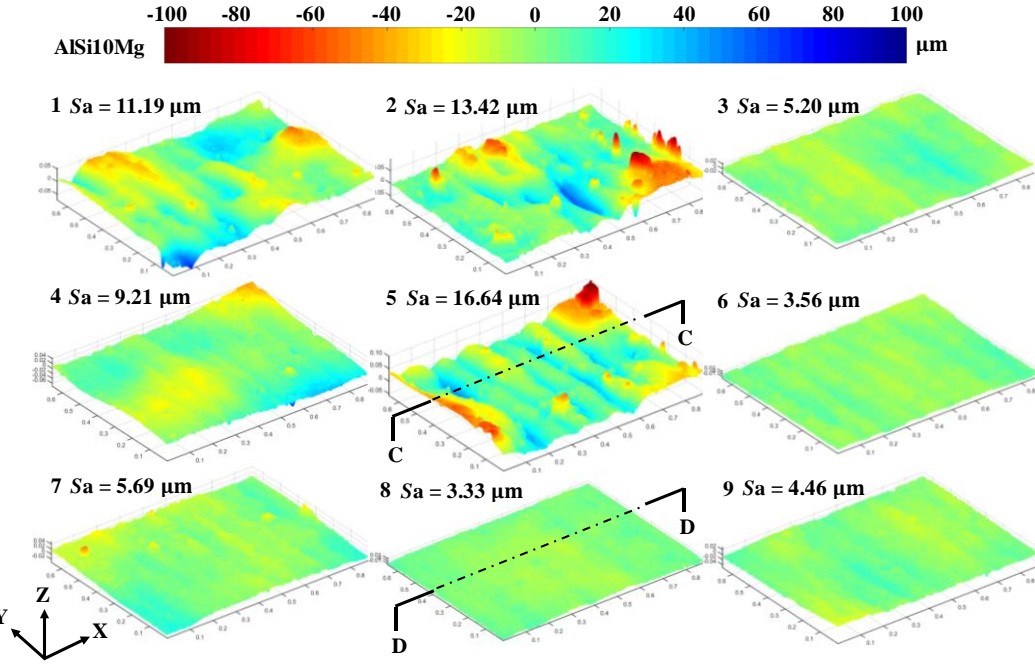

**Figure 5.** Three-dimensional contours of the AlSi10Mg sample after laser polishing.

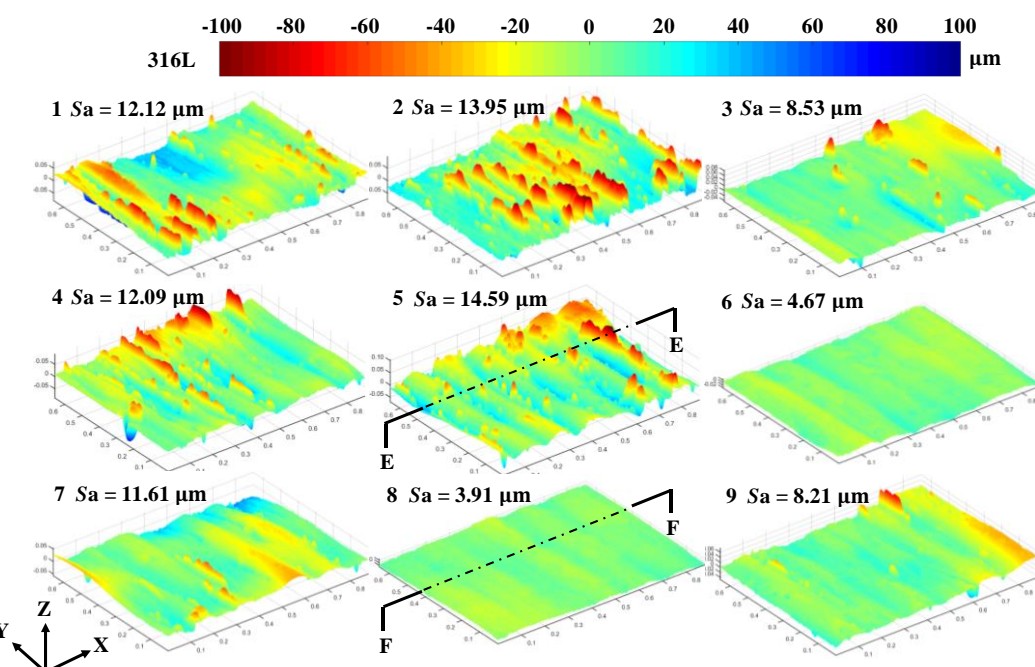

**Figure 6.** Three-dimensional contours of the 316L sample after laser polishing.

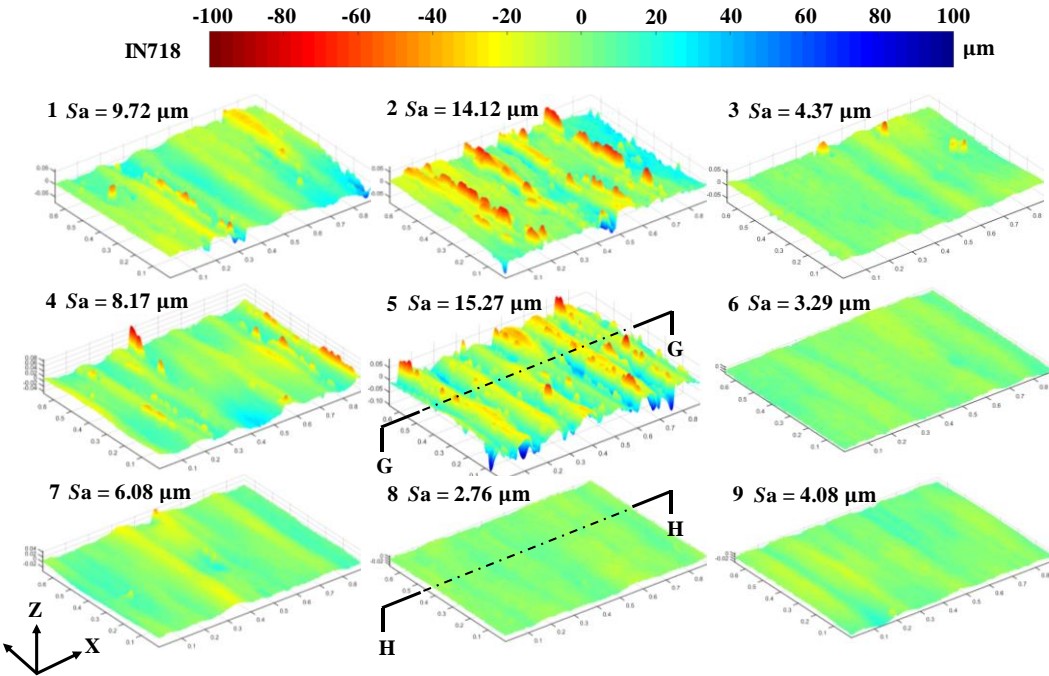

**Figure 7.** Three-dimensional contours of the IN718 sample after laser polishing.

Figures 4–7 show the morphologies of the four sample surfaces after laser polishing, with the polishing strategies listed in Table 2. For the Ti6Al4V alloy, the roughness values obtained with the eight laser-polishing strategies clearly differ considerably, ranging from 3.32 μm to 8.14 μm. It is obvious that the surfaces of the sample after laser polishing are smoother than the initial surface. For the AlSi10Mg alloy, the roughness values for each laser-polishing strategy range from 3.33 μm to 13.42 μm. For the 316L alloy, the roughness values for each laser-polishing strategy range from 3.91 μm to 13.95 μm. For the IN718 alloy, the roughness values for each laser-polishing strategy range from 2.76 μm to 14.12 μm. By comparing the three-dimensional contours of the four kinds of alloy materials, it was found that, for the given laser-polishing parameters (laser power and scanning speed), the rules of polishing effect are the same regardless of the material investigated in this work.

The initial surface roughness of the four alloys differs greatly, this being related to their powder characteristics, such as the sphericity and flowability. From Figure 1, it can be seen that the Ti6Al4V alloy powder has a higher level of particle sphericity than the other three alloys, which is linked to the flowability of the power. Thus, the uniformity of the layer is affected, as well as the surface roughness of the final parts. Although the initial surface roughness values of each alloy are different, after laser polishing, all of the material surfaces tend to be smooth, and the final surface roughness is about 3 μm.

The line profiles of the processed surfaces are shown in Figure 8. They were acquired along the centreline of the polished surfaces shown in Figures 4–7. Figure 8 shows the best polishing parameters for the four typical alloy samples and the morphology curves obtained with those parameters. The best polishing parameters are a laser power of 400 W and a scanning speed of 0.5 m/s, which gives the largest energy density. It is obvious that the surface profile after polishing is much smoother than that for the initial surface. The larger peaks and valleys on the initial surface have basically disappeared, and the surface profile is smooth. Although the roughness of the initial surfaces of the four samples are not uniform, the surface roughness of the surfaces of all four materials has been significantly reduced after laser polishing. For the Ti6Al4V alloy, the initial peak–valley value is more than 70 μm, and it is reduced to about 10 μm after laser polishing. For the AlSi10Mg, 316L and IN718 alloys, their largest peak–valley values decrease from 45, 48 and 77 μm, respectively, to about 10 μm.

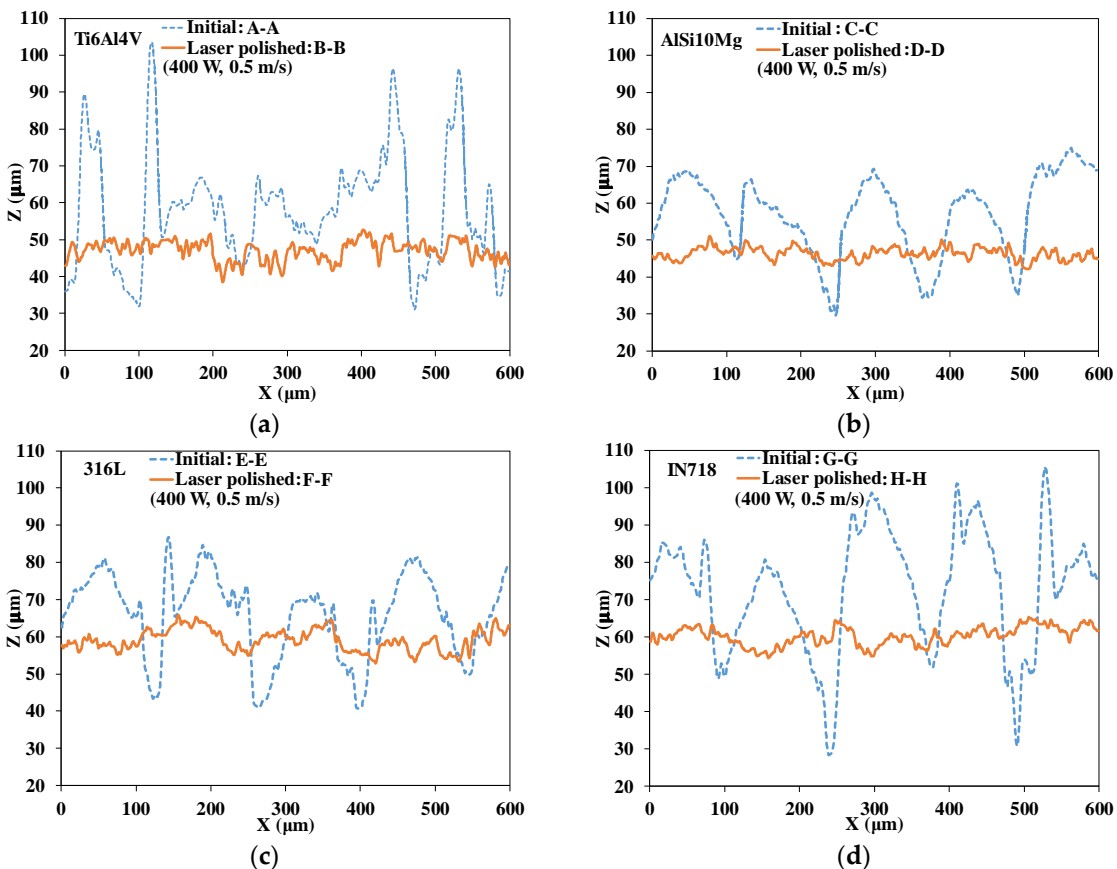

**Figure 8.** Data plot of line profiles extracted from three-dimensional contours: (**a**) Ti6Al4V, (**b**) AlSi10Mg, (**c**) 316L and (**d**) IN718.

## 3.2. Influence of Laser Power and Scanning Speed

Figure 9 shows the relationship between the laser power, scanning speed and surface roughness ($S_a$) values. It is obvious that, after laser polishing, the surface roughness of the materials will be reduced. When the laser scanning speed remains constant, the surface roughness decreases with an increase in the laser power. Meanwhile, when the laser power remains constant, a lower roughness value is obtained with a smaller scanning speed. When the laser powers are 300 W and 400 W, the influence of the scanning speed on the surface roughness is more obvious.

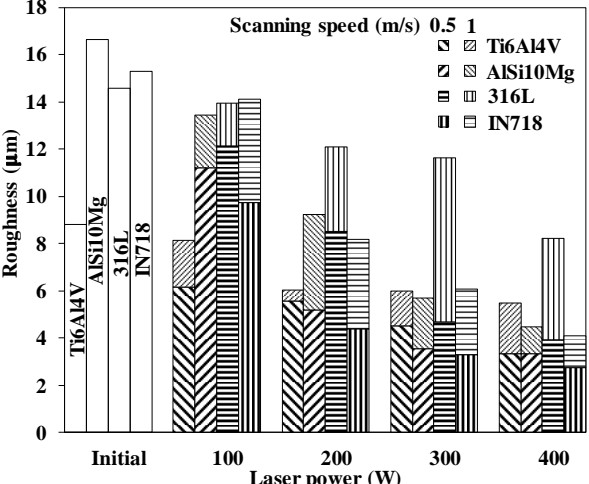

**Figure 9.** Relationship between laser power, scanning speed and roughness.

When the scanning speed is 1 m/s, the surface roughness is not significantly affected by the smaller laser-polishing power, especially at 100 W. This is mainly due to the fact that the small energy input only partially melts the material surface, resulting in a smaller molten pool, which is not conducive to the wettability of the materials in the adjacent areas. After solidification, there are obvious scanning marks on the surface, and the roughness exhibits very little change. Owing to the decrease of the laser moving speed, a scanning speed of 0.5 m/s produces a molten pool that is able to wet the adjacent area so that a larger molten pool can be obtained, producing a smoother surface after solidification. Similarly, when the laser scanning speed is constant, an increase in the laser power causes the molten pool to increase in size. Therefore, a smoother surface with a smaller roughness value can be obtained. For all four materials, the polishing effect is optimum when the laser speed is 0.5 m/s and the laser power is 400 W.

### 3.3. Influence of Laser Energy Density

Figure 10 clearly shows that the reduction in the surface roughness is related to the energy density of the polishing laser, as determined by the laser power, scanning speed and hatch space. As the energy density increases, the effect of laser polishing becomes more obvious. The low LED is not sufficient to penetrate the sample surface, causing that the metal surface melts incompletely and the molten pool diffuses inadequately. In this case, the solidification time is transient, and the polishing effect is not obvious. With the increase of energy density, the solidification time becomes sufficiently long to make the surface molten pool and surrounding material wet and diffused sufficiently. Under the influence of gravity and surface tension, the molten pool flows, and the surface is redistributed. The larger the energy density, the larger the molten pool, and the internal flow becomes more intense, causing the resulting surface to be smoother after solidification. In the present study, the surface roughness was about 3 μm with the best polishing parameters.

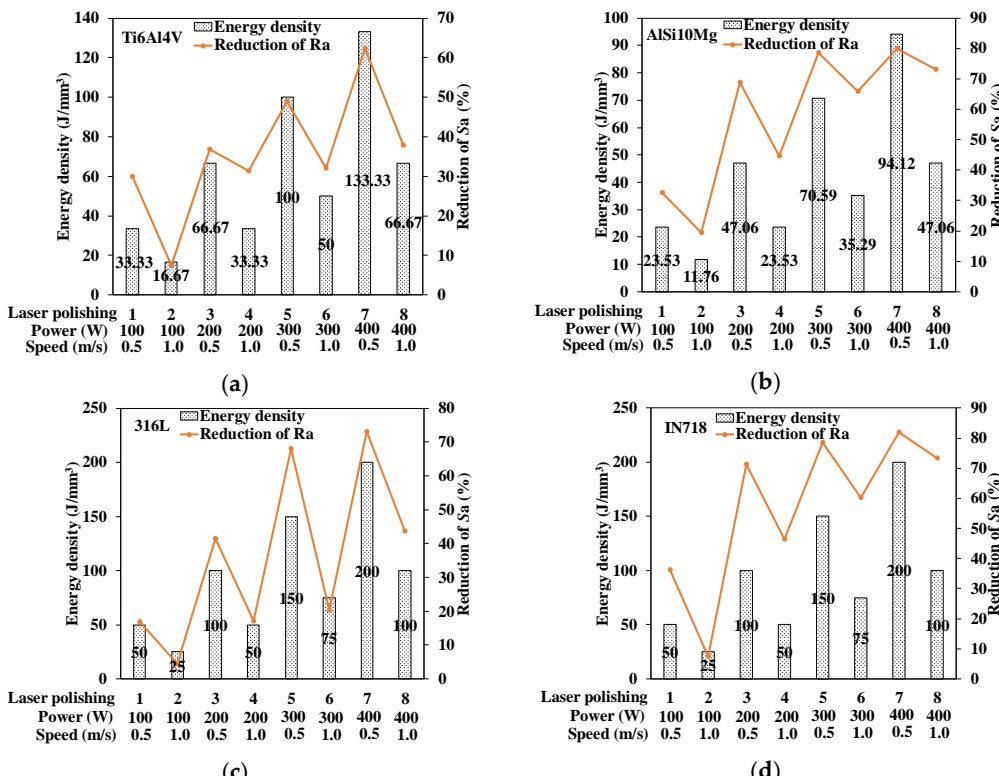

**Figure 10.** Relationship between energy density and roughness reduction rate: (**a**) Ti6Al4V, (**b**) AlSi10Mg, (**c**) 316L and (**d**) IN718.

Meanwhile, for any given energy density, the laser power and scanning speed will vary ($P = 100$ W, $v = 0.5$ m/s; $P = 200$ W, $v = 1$ m/s; $P = 200$ W, $v = 0.5$ m/s; and $P = 400$ W, $v = 1$ m/s). At this time, the results of laser polishing will also vary slightly. As the laser power is increased, the polishing effect often becomes more obvious such that the surface roughness decreases with an increase in the laser power. That is to say, for a given energy density, the effect of the laser power is more important than the scanning speed.

## 4. Conclusions

Using selective laser melting technology, laser polishing of four kinds of alloy materials commonly used in metal additive manufacturing was studied using eight different process parameters. The main conclusions can be summarized as follows:

(1) Laser polishing of SLM samples can be carried out by adjusting the laser parameters, without adding any additional lasers. That is to say, laser polishing and printing can be done using the same device. This largely prevents oxidation during the laser-polishing process.

(2) Laser polishing of four typical alloys with eight laser strategies was carried out by using the printing laser. By changing the laser power and scanning speed, surfaces treated with different laser energy densities were obtained.

(3) Observations of the top surface morphology revealed that the polishing effect was obvious and the top surface roughness was greatly reduced. For the Ti6Al4V, AlSi10Mg, 316L and IN718 alloys, the $S_a$ decreased by 62.3%, 80.0%, 73.2% and 81.9%, respectively.

(4) The laser polishing mechanism relies on melting and then re-solidification of metals. When the laser irradiates the surface of the material, the material melts to form a molten pool, with evaporation occurring from the local temperature to boiling point. The surface molten pool and surrounding material undergo sufficient wetting and diffusion, causing the crests and troughs of the initial surface to become smoother.

**Author Contributions:** D.Z. designed the experimental scheme, completed the samples preparation and wrote the initial draft of the manuscript. J.Y. participated in the samples preparation. H.L. supervised this work. S.S. was involved in manuscript preparation. X.Z. participated in the experimental scheme. C.Z. provided guidance for experiments. C.S. was involved in the roughness test experiment. L.L. participated in the data analysis. C.D. checked of accuracy of whole manuscripts. All authors have read and agreed to the published version of the manuscript.

**Funding:** This work is supported by the National Key Research and Development Program of China (grant no. 2017YFB1103900, 2018YFB1105100), the Key Research and Development Program of Guangdong Province (grant no. 2018B090905001), the Natural Science Foundation of Guangdong Province (grant no. 2018A030313044), the Science and Technology Planning Program of Shenzhen (grant no. JCYJ20170816171733384), the Fundamental Research Funds for the Central Universities (grant no. 2042018kf0240) and the Hubei Provincial Natural Science Foundation of China (grant no. 2017CFB657).

**Acknowledgments:** The authors would like to thank Wuhan Huake 3D Technology Co. Ltd. Wuhan, China for providing the technical support.

**Conflicts of Interest:** The authors declare no conflict of interest.

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
