# Peer review of "Investigation of Laser Polishing of Four Selective Laser Melting Alloy Samples"

_applsci, doi:10.3390/app10030760_

Round 1

Reviewer 1 Report

Dear authors

1- The English language is very poor and must be revised and corrected in addition to the use of the proper scientific terms.

2- There are many errors which cannot be placed in the field so I have put them on the article PDF file. Please check and adjust accordingly.

3- Please do not use (we will...). No we's and/or I's can be used in an article.

4- Clearly, this study was based on one single test, no repeatibility test was carried out. Scientific facts cannot be based on one test results and that the CI's and/or standard deviation must be presented.

Regards

Reviewer 2 Report

General Comments:

The authors are reporting an in-situ laser polishing post-processing method for reducing the surface roughness of selective laser-melted samples. The authors have investigated the applicability of the proposed laser polishing method on four different materials which are commonly used in powder bed fusion additive manufacturing, namely Ti6Al4V, AlSi10Mg, 316L and IN718.

Even though the paper reports on an interesting topic, this method has already been reported in literature and the paper lacks novelty. Also, authors should have commented on the effect of the laser polishing on the material properties of the laser polished samples, since this is a critical point for applying the method in industrial environment. Finally, the authors should have also comment how the method is applicable to any other geometries than a flat plane.

Taking into considerations the above points, I cannot recommend the paper for acceptance to the Applied Science at its current state. Thus, I would recommend a major revision at this time. Some more specific comments are provided below:

Specific comments:

Line 158: Authors have calculated surface roughness using equation 2 and 3, but there is no mention if they have applied form removal operations and what is the cutoff wavelength use to calculate the surface roughness parameters?

Line 165: Authors have reported only Ra in regards to surface morphology, but there are another 23 standardized surface roughness parameters, which probably could explain better the improvement in surface  roughness (Rq,Rz, Rt, Rvm Rsk, Rku and etc.). Have the authors looked at those parameters as well? Also, is not areal based parameters (Sa, Sq an etc.) better way to describe the surface morphology of additive manufacturing parts due to the lack of directional surface properties?

Line 198: was there any material ablation apart for the polishing effect?

Line 205: Authors are saying: “In the following study, we will optimize polishing strategy by simulation to further reduce surface roughness.”- This does not convey any useful information to the reader of this paper.

Line 271: Conclusion (3): What is the reason for having different surface roughness improvements on the four different materials?

Reviewer 3 Report

The manuscript reads well. The results are presented comprehensive.

I suggest the following minor revisions:

1) Please use the ISO definitions instead of "3D-printing" colloquial language.

2) Figure 1 and 3 should be reworked with respect to readybility of the legends.

3) At least building chamber dependence should be dicussed with respect to the experimental setup, since the position has a significant influence is present here for global geometries. See for example in the MDPI Technologies journal "Hartmann C, Lechner P, Himmel B, Krieger Y, Lueth TC, Volk W. Compensation for Geometrical Deviations in Additive Manufacturing. Technologies. 2019; 7(4):83."

I suggest to also contrast micro and macro levels of error, for example, in the Introduction section.

Reviewer 4 Report

In this article different laser polishing processes based on different laser power and marking speed were tested on four different alloy. Here you can find some comments:

Overall comment:

1) The authors should better specify what is the novelty of this research? As they know the literature is full of similar researches so is better to highlight what is the effort of this article repect to the state of art

2) The analisys of results must be revised. First of all how many replica of Ra have been recorded? Why Ra and not Sa that is nowadays a best parameter to measure roughness reduction? Why the authors do not follow a robust statistical approach (as the analisys of variance as example) to analize the results?

3) English must be strongly revised, there are different sentences totally wrong; the title too seems to be wrong

Minor Chapter comments:

Introduction: the authors add a lot of citation about the effect of SLM parameters on the surface quality and too less about the laser polishing, it is better to reverse and improve literature about polishing on SLM parts that is the research topic of the authors

Material and Methods: line 145: please justify why the hatch spaces was reduced

Results: see the overall comment 2

Round 2

Reviewer 1 Report

Dear Authors,

Thanks for the responds and improvements you did which clearly enhance the article in general. I would recommend the following minor adjustment to be carried out. The article can be then published without any following revision. Most of the following comments are about to explain the process/reasons/strategy to the reader NOT me.

1- I asked: Did you do any optimization? if so, what was the result?

You answered: Optimized manufacturing parameters of the SLM equipment were used in sample printing.

This is clear, we need to explain why did you select these processing parameters with these levels as long as no optimisation was carried out.

2- My comment: This is so far, a top surface polishing of the part. Does this suit the sides, internal geometries? please explain

You answered: In our work, we focus on the top surface polishing of samples. The sides and internal geometries do not been taken into consideration. 

Again this is clear for me and you, but we need to put this as an additional bullet point in the conclusion as one limitation of the process. 

Thanks and best regards.

Reviewer 2 Report

Accept at its current form
